# CloudNFMM: A Hybrid Hierarchical and Local Neural Operator Inspired by the Fast Multipole Method

## Abstract

The Fast Multipole Method (FMM) is an efficient numerical algorithm used to calculate long-range forces in many-body problems, leveraging hierarchical data structures and series expansions. In this work, we present the Cloud Neural FMM (CloudNFMM), a new neural operator architecture that integrates the hierarchical structure of the FMM to learn the Green's operator of elliptic PDEs on point cloud data. The architecture efficiently learns representations for both local and far-field interactions. The core innovation is the local attention, a specialised local attention mechanism which models complex dependencies within a small neighbourhood of points. We demonstrate the effectiveness of this approach, and discuss possible extensions and modifications to the CloudNFMM architecture.

## 1 Introduction

Solving partial differential equations (PDEs) is fundamental to countless fields in science and engineering. While traditional numerical solvers are highly refined, they can be either brittle – needing to be tuned for each new problem – or computationally expensive for large computational domains. This has spurred the development of deep learning-based methods, particularly neural operators, which aim to learn the underlying solution operator mapping from input parameters to the solution function. Among these, the **F**ourier **N**eural **O**perator (**FNO**) (Kovachki et al., 2021a) has emerged as a state-of-the-art architecture, demonstrating remarkable success by performing convolutions in the frequency domain. However, a significant limitation of the FNO is its reliance on the Fast Fourier Transform (FFT), which constrains it to data structured on uniform, regular grids.

Many real-world problems are defined on irregular domains or are naturally represented by unstructured data, such as point clouds or meshes. To address this, architectures based on Graph Neural Networks (GNNs) and Transformers (Vaswani et al., 2017) have been proposed. However, neural operators based on these methods introduce significant bottlenecks for large-scale simulations; either requiring many message-passing steps or by relying on global attention mechanisms, incurring a $O(N^2)$ computational cost.

To overcome these challenges, we draw inspiration from a very successful class of direct solver algorithms: FMM. The FMM is designed to compute long-range interactions in $N$-Body problems with near-linear complexity, typically $O(N)$ or $O(N \log(N))$. This is achieved by hierarchically decomposing the computational domain and using low-rank approximations of the interaction kernel for far-field interactions. This principle of separating local and global computations provides a blueprint for a more scalable and geometrically robust neural operator architecture.

**Contribution** We introduce the CloudNFMM, a neural operator which uses the hierarchical structure of the FMM for global interactions with a local attention mechanism for local interactions. The proposed architecture operates directly on unstructured points (point clouds), making it resolution-invariant and freeing it from the grid-based constraints of methods like the FNO. We demonstrate the efficacy of the CloudNFMM on a variety of time-harmonic PDE problems, showing that our model achieves performance that is superior to, or on par with, other established neural operator architectures at a fraction of the parameter count.

## 2 BACKGROUND

### 2.1 FAST MULTIPOLE METHOD

We will outline a high-level discussion of the FMM's information flow. For a more detailed discussion of the FMM, see Appendix A. The FMM (Rokhlin, 1985) designed to efficiently compute long-range forces in $N$-Body problems utilising both a *low-rank approximation* of the kernel, $G(x, y)$, and a *hierarchical decomposition* of the domain $D$ via a *quad-tree* in 2D (or an *oct-tree* in 3D).

The FMM decomposes the domain $D$ into $4^L$ disjoint boxes (in 2D, it is $8^L$ in 3D), $\beta_i$, where $L$ is the depth of the FMM's quad-tree. For all $\beta_\tau$, we partition the domain into its far-field, $\mathcal{F}_\tau$, and near-field, $\mathcal{N}_\tau$, neighbours. A source-box $\beta_\sigma$ belongs to the near-field $\mathcal{N}_\tau$ of a target-box $\beta_\tau$ if we define the far-field, $\mathcal{F}_\tau$, and near-field, $\mathcal{N}_\tau$, of $\beta_\tau$ respectively, furthermore, we denote the sub-scripts $\tau$ and $\sigma$ to indicate a given target box and source box respectively. Boxes belong to $\mathcal{F}_\tau$ if $\beta_\sigma$ – with $c_\tau$ and $c_\sigma$ being the centre of each respective box – satisfies $2b_l \leq |c_\tau - c_\sigma|$ with $b_l$ being the length of a box at a given level $l$.

**Upwards Pass**: The upwards pass aggregates information from the source points up the tree using the following operators:
$\mathbf{T}_\sigma^{\text{ofs}}$ – this computes the compact representation vector $\mathbf{q}_\sigma$ for each box on the leaf level.
$\mathbf{T}_{\Sigma,\sigma}^{\text{ofo}}$ – shifts the $\mathbf{q}_\sigma$ between the tree's levels towards level 2 of the tree[1], creating the outgoing potential tensors[2] for each level $l$; $\mathbf{Q}_l$. This is done to the parent box $\mathbf{q}_\Sigma$ from the children boxes, $\mathbf{q}_\sigma \in \mathcal{C}_\Sigma$. This operation is described mathematically in equation 9 and equation 10.

**Downward Pass**: The downward pass gathers the vectors outgoing potentials tensors $\mathbf{q}_\sigma \in \mathbf{Q}_l$ corresponding to boxes $\beta_\sigma \in \mathcal{F}_\tau$ for each $\tau \in \mathbf{Q}_l$ using the following operators:
$\mathbf{T}_{\tau,\sigma}^{\text{ifo}}$ – this computes the incoming vector $\mathbf{h}_\sigma$ from the outgoing vector $\mathbf{q}_\sigma$.
$\mathbf{T}_{\tau,T}^{\text{ifi}}$ – shifts $\mathbf{h}_\tau$ between the tree's levels from level 2 to the leaves, going from a parent box $\beta_T$ to $\beta_\tau \in \mathcal{C}_T$. This operation is described mathematically in equation 11.

**Leaf Pass**: The leaf pass computes the potential at a point $x$ from the contributions from both $\mathcal{N}_\tau$ and $\mathcal{F}_\tau$ using the following operators:
$\mathbf{T}_\tau^{\text{tfi}}$ – evaluates the potential $\mathbf{h}_\tau$ at all the points $x_i \in \beta_\tau$, this is the contribution to the potential from the points $x_j \in \mathcal{F}_\tau$.
$G(x, y_j)$ – is the kernel, which directly evaluates the potential between $x$ and all the points $y_j \in \mathcal{N}_\tau$. This operation is described mathematically in equation 1:

$$v(x) = \overbrace{\sum_{y_j \in \mathcal{N}_\tau \setminus \{x\}} G(x, y_j) f(y_j)}^{\text{Near Field Contribution}} + \overbrace{\mathbf{T}_\tau^{\text{tfi}}(x; \mathbf{h}_\tau)}^{\text{Far Field Contribution}} \tag{1}$$

Collectively, the upward and downward passes constitute the *tree-level* operations – they are responsible for communicating information across the domain by aggregating sources into compact representations (up the tree) and propagating their far-field influence back down (down the tree). The *leaf-level* operations – described in the leaf pass – involve direct kernel evaluation in a local neighbourhood. This separation of long-range (tree) and short-range (leaf) computations is the core principle we adapt in our architecture.

**Requirements:** To achieve resolution independence for the NFMM, we need to:
Firstly remove the dependence of the NFMM on requiring that data is on a uniform grid[3].
Secondly, reformulate the NFMM's local interactions to learn a local interaction kernel[4], and have the information flow as the near-field contribution in equation 1.

---

[1] As the spatial resolution in the higher levels, levels 0 and 1, is too coarse to allow for separation of the near-field and far-field.

[2] Note that each $\mathbf{q}_\sigma \in \mathbf{Q}_l$ corresponds to a box $\beta_\sigma$ on level $l$ of the quad-tree.

[3] This would require that the $\mathbf{T}^{\text{ofs}}$, $\mathbf{T}^{\text{tfi}}$, and $\mathbf{G}_\theta$ operators are can handle variable sized inputs.

[4] With the following requirements: One, no self interactions for sources/targets in $\beta_\tau$. Two, that we only compute the contribution to the points in $\beta_\tau$, from the points in $\mathcal{N}_\tau$, not the other way around.

## 2.2 Neural Operator

Neural operators (Kovachki et al., 2021a; 2024), aim to learn operators between different function spaces with some light conditions on their domains. Originally outlined by Kovachki et al. (2021a), and further formalised by Berner et al. (2025), neural operators have two key properties. One, neural operators should be discretisation-agnostic[5]. Two, they should have a fixed number of parameters for every discretisation.

Neural operators were inspired by the DeepONet (Lu et al., 2021), neural operators are fashioned after a traditional deep learning architecture, where for each layer $t$ contains a linear operation with a bias followed by a non-linearity. In the neural operators framework, there are 3 major components: the lifting operator, $\mathcal{P}$, the blocks, $\{\mathcal{B}_t\}_{t=1}^T$, and the projection operator $\mathcal{Q}$. Note that $\mathcal{P}$ and $\mathcal{Q}$ are channel wise and only $\mathcal{B}_t$ operate along the spatial domain. There are $T$ blocks, with each block containing: a channel-wise linear layer, $W_t$, and spatial kernel operator, $\mathcal{K}_t$, followed by a non-linearity, $\sigma_t$,

$$\tilde{\mathcal{L}}_\theta = \mathcal{Q} \circ \sigma_T \underbrace{(W_T + \mathcal{K}_T + b_T)}_{\text{Block } T} \circ \cdots \circ \sigma_1 \underbrace{(W_1 + \mathcal{K}_1 + b_1)}_{\text{Block } 1} \circ \mathcal{P}. \tag{2}$$

Let $v_t$ be our solution at our current step and $\kappa^{(t)}$ be our learnt integral kernel at a layer $t$, which may depend on $(x, y, a(x), a(y), v_t(x), v_t(y))$. To mathematically define our spatial kernel operator – dubbed the 'non-local' operator – we represent $\mathcal{K}_t$ as an integral. For some measure, $d\nu_t(dy)$, on the domain of integration $D$, we define $\mathcal{K}_t$ in terms of $\kappa^{(t)}$ as follows:

$$(\mathcal{K}_t(v_t))(x) = \int_{D_t} \kappa^{(t)}(x, y, a, v_t) v_t(y) \, d\nu_t(dy). \tag{3}$$

Depending on the class of problems and type of kernel we aim to learn, the structure of the kernel, $\kappa^{(t)}$, and the computation of the integral transform in equation 3 can be simplified, giving rise to different architectures. Boullé & Townsend (2023), viewing operator learning through the lens of linear algebra, outline four main approaches: these are the 'Graph neural operator' (GNO), 'Low-rank neural operator', 'Multipole Graph neural operator', and the FNO.

## 2.3 Related work

**Neural Fast Multipole Method**   We propose the **N**eural **F**ast **M**ultipole **M**ethod (**NFMM**), a novel architecture that integrates the hierarchical information flow of the classical FMM into a neural operator framework for learning the Green's operator of elliptic PDEs. The core idea is to replace the FMM's traditional, handcrafted translation operators, which depend on an analytically available Green's kernel, with MLPs. This approach preserves the FMM's efficient partitioning of near and far-field interactions, including its characteristic upward and downward passes through a hierarchical tree structure, while circumventing the need for a-priori knowledge of the interaction kernel. For a detailed breakdown of the NFMM architecture, we refer the reader to Appendix B. Our present work is expanding the NFMM to be discretisation-agnostic – much like the original FMM.

**Multipole Graph Neural Operator:**   The **M**ultipole **G**raph **N**eural **O**perator (**MgNO**) (Li et al., 2020) is a model that merges concepts from graph neural operators and low-rank neural operators to efficiently learn PDE solution operators. Inspired by the FMM and $\mathcal{H}^2$-matrices, it uses a message-passing algorithm called a V-Cycle on hierarchical graphs to enforce a low-rank structure on the interaction kernel, particularly for long-range components. This architecture functions as an iterative solver, with the final learned kernel resembling a hierarchical $\mathcal{H}$-matrix (Martinsson, 2019). The MgNO differs significantly from the NFMM; while both draw inspiration from the FMM, the NFMM is a more direct adaptation that replaces the FMM's handcrafted operators with learnable MLPs, while explicitly preserving the upward and downward pass structure. In contrast, the MgNO employs a more generalised graph-based V-Cycle for its message passing, focusing on kernel decomposition rather than adapting the FMM's information flow.

---

[5]This is typically loosened to being resolution-agnostic for architectures such as the FNO.

**Graph Neural Operators:** Work on learning physics simulations directly on meshes has been significantly advanced by modern deep learning architectures. The MeshGraphNets (Pfaff et al., 2020) architecture is a Graph Neural Network (GNN) using an Encode-Process-Decode structure. Its key innovation is a dual message-passing scheme that operates in two distinct spaces: mesh-space, using the mesh's connectivity to model internal dynamics like material properties, and world-space, using proximity-based edges to capture external interactions such as collisions. Building on this, **EAGLE** (Janny et al., 2023) addresses the challenge of modelling more complex, unsteady turbulent flows and the inefficiency of iterative message passing for capturing long-range dependencies using a novel mesh transformer architecture. To overcome the quadratic complexity of attention on large meshes, the model first performs geometric clustering and learned graph pooling to create a coarser representation of the mesh, then applies multi-head self-attention on the expressive cluster embeddings. This allows the model to integrate global information and capture long-range interactions, such as airflow patterns, in a single step, outperforming iterative GNNs like MeshGraphNet on complex benchmarks. Similarly, the CloudNFMM also avoids the quadratic-complexity of transformers by solving the global solve on a coarse-grid, however, EAGLE only uses attention on the coarse graph and uses a decoder to update the fine global mesh.

**Transformer Neural Operators:** Recent advancements in neural operators have focused on overcoming the geometric and discretisation limitations of earlier models for solving PDEs, leveraging the improvements in transformer implementation and theory. The **Geo**metry-aware **F**ourier **N**eural **O**perator (Geo-FNO) (Li et al., 2022b) addresses a key constraint of the popular FNO, which is its reliance on uniform rectangular grids due to its use of the Fast Fourier Transform (FFT). The Geo-FNO introduces a framework that learns a diffeomorphic deformation to map from an irregular domain into a regular domain where the FNO can be efficiently applied, before the result is mapped back to the irregular domain. The **O**perator Trans**former** (OFormer) (Li et al., 2022a) proposes an attention-based architecture that makes few assumptions about the input grid structure. It leverages self and cross-attention to function as a learnable integral operator, with the cross-attention mechanism decoupling the input and output domains to allow for queries at arbitrary locations. For time-dependent problems, the OFormer employs a recurrent MLP to propagate the system's dynamics efficiently in the latent space. The grap**H** transfor**M**er neur**A**l op**E**ra**T**or (**HAMLET**) (Bryutkin et al., 2024) is the first neural operator framework to employ a graph transformer for solving PDEs. HAMLET constructs a graph from the input data and uses graph transformer blocks for encoding, a cross-attention operator for integrating query locations, and a similar recurrent MLP as OFormer for time-dependent PDEs. These neural operators are similar to the CloudNFMM, using a transformer-based architecture to learn an integral operator. However, the CloudNFMM differs from these approaches by splitting the computational domain into long-range and local interactions, as opposed to using an attention-mechanism between all points in the domain.

## 3 METHOD

### 3.1 CLOUD NFMM

In the CloudNFMM, we expand upon the original NFMM – outlined in Appendix B – by reworking the $\mathbf{T}^{\text{ofs}}$, $\mathbf{T}^{\text{tfi}}$, and $\mathbf{A}$ operators; although we will denote $\mathbf{A}$ as $\mathbf{G}_\theta$ in this work. The driving motivation behind this work is to replace the $\mathcal{K}_t$ from equation 2 with an operator which models the information flow of the FMM. As the FMM is a hierarchical algorithm, our FMM-inspired neural operator is also a hierarchal algorithm; having both a *tree pass* and a *leaf pass*. The tree pass is handled by the original version of the NFMM and was constructed by simply replacing each FMM operator with either a linear layer, or a $2-$layer MLP. The leaf pass is the focus of this work, and is implemented via the use of a spatially local attention mechanism. In order to exchange information between the leaf level and the tree level, we also need to rework the $\mathbf{T}^{\text{ofs}}$ and $\mathbf{T}^{\text{tfi}}$ operators from the original NFMM to meet the requirements outlined above.

**Data Preprocessing** To enable the hierarchical structure of the NFMM for point-based data, the input domain is first partitioned into a uniform grid of square cells, referred to as boxes, similar to the patches used by a Vision Transformer (Dosovitskiy et al., 2020). Our input data, consisting of $\{\mathbf{x_i}\}_{i=1}^N$ points represented as a tensor of feature vectors $\mathbf{f}_i \in \mathbb{R}^{d_L}$, with the pre-processing partitioning assigning each point $\mathbf{x}_i$ to a specific patch. As this architecture should be resolution

invariant, in general there will not be a constant number of points within each patch. To create a tensor with full dimensions, each patch's point list is processed to have a fixed size, $N_b$, which is the maximum number of points in a given patch[6]. The output of this preprocessing step is a structured tensor of shape $[B, M, N_b, d_L]$, where $B$ is the batch size, $M$ is the number of boxes, $N_b$ is the (maximum) number of points in each box, and $d_L$ is the dimension of the leaf feature space. This format is crucial for the hybrid local and hierarchical structure of the NFMM, as the hierarchical FMM operator works on the box grid while the local operator works on the points in the spatial boxes.

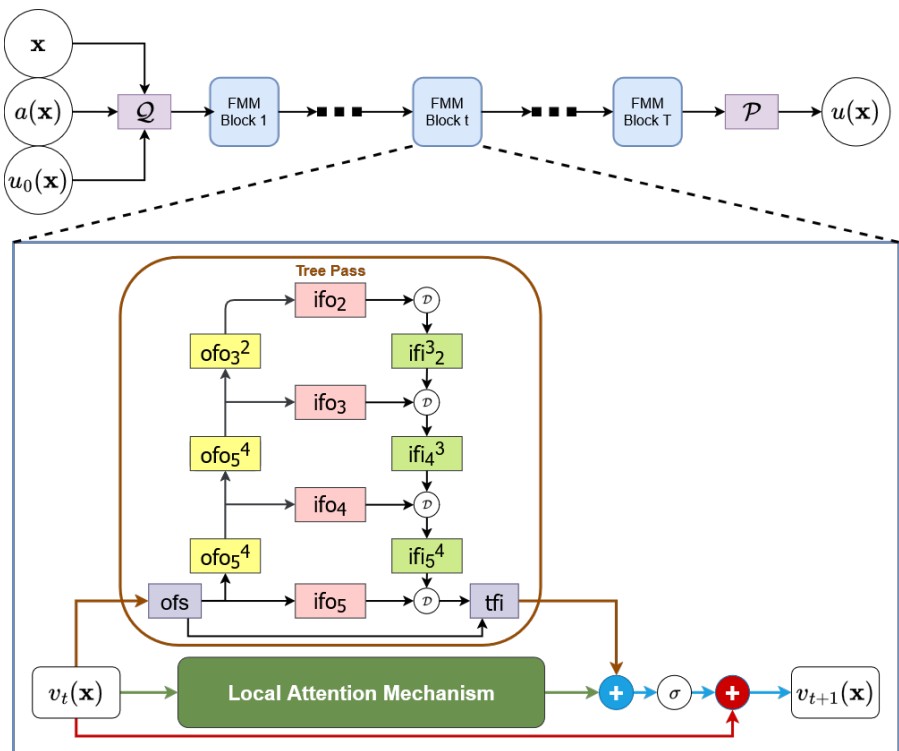

Figure 1: The architecture of the CloudNFMM neural operator.

## 3.2 CLOUD NFMM COMPONENTS

### 3.2.1 OFS OPERATOR

The $\mathbf{T}^{\text{ofs}}$ operator is responsible for the crucial aggregation step in the upward pass of the NFMM. Its primary purpose is to compress the rich information contained within a set of points in a single box into a compact, representative feature vector for the box. This is achieved by first lifting the leaf-level features – $\mathbf{x}_i \in \mathbb{R}^{d_L}$ – into a higher-dimensional – $\mathbf{q}_i \in \mathbb{R}^{d_T}$ – space using a MLP. The representative vector – $\mathbf{q}_\sigma$ – for each box is then computed as a magnitude-weighted centre of mass of these lifted features. The magnitude of each lifted feature vector is used to determine the relative contribution of each point to the final aggregated vector. This approach – outlined in equation 4 – is similar to a soft aggregation, allows points with more significant or prominent features to have a greater influence on the final aggregated vector.

### 3.2.2 TFI OPERATOR

The $\mathbf{T}^{\text{tfi}}$ operator is the final stage of the downward pass, translating the coarse-level FMM approximations into point-level updates. It takes the aggregated feature vector from a parent box and uses it

---

[6]The value of $N_b$ only needs to be done per example or batch, however to speed up training we process all examples in a dataset to have $n_b$ points.

to update the individual point representations within each of its child boxes. This is achieved by an inverse-weighted update mechanism, which is outlined in equation 5. First, the operator calculates the distance between each individual point's feature vector and the incoming representative vector in the high-dimensional space $- d_i = \|\mathbf{q}_i - \mathbf{h}_\tau\|_2$. This distance is then used to compute an interaction weight, using an inverse weighting scheme, which is then normalised. The point's features are then updated by 'pulling' them toward the incoming vector, with the strength of this pull determined by the calculated weight. The updated features, $\tilde{\mathbf{q}}_i$, now incorporating information from the parent box, are projected back to the original dimension using a MLP,

$$\mathbf{q}_\sigma = \sum_{i \in \beta_\sigma} \frac{\|\mathbf{q}_i\|_2 \cdot \mathbf{q}_i}{\sum_{i \in \beta_\sigma} \|\mathbf{q}_i\|_2}, \qquad (4) \qquad \tilde{\mathbf{q}}_i = \mathbf{q}_i + \frac{\|\mathbf{q}_i - \mathbf{h}_\tau\|_2^{-1} \cdot (\mathbf{h}_\tau - \mathbf{q}_i)}{\sum_{i \in \beta_\tau} \|\mathbf{q}_i - \mathbf{h}_\tau\|_2^{-1}}. \qquad (5)$$

### 3.2.3 LOCAL ATTENTION OPERATOR

In order to satisfy the requirements outlined above, the only architectures available were either GNNs, transformer-based architectures, and state-space architectures. Latent-space models were not considered for $A$ due to the fixed state dimension found within these models[7], leaving GNNs and transformers as possible architectures to build the $\mathbf{G}_\theta$ operator. We note that transformers are a special class of message passing neural networks – this is outlined in Appendix C, with a similar discussion seen within (Bryutkin et al., 2024) – thus, we have focused on transformers due to existing efficient implementations and as we do not need to construct graphs due to the patched nature of the data.

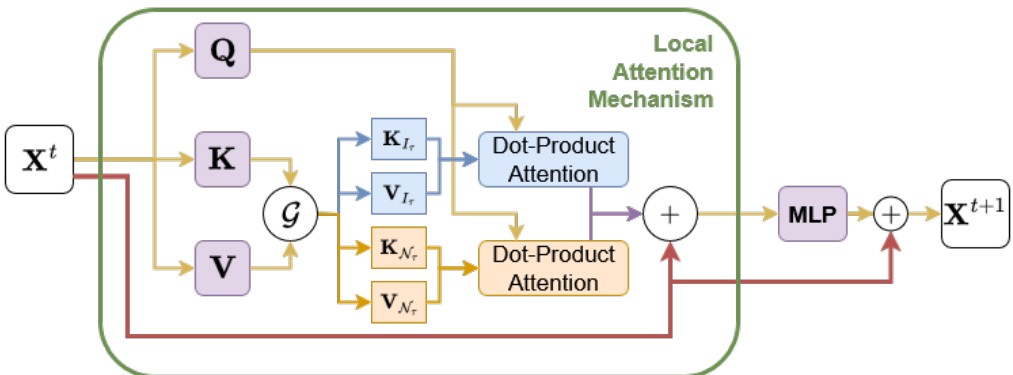

Figure 2: The architecture of the Local Attention Operator, where $\mathcal{G}$ is a gathering operation for the local boxes.

The algorithm represented in Figure 2 is a shared attention mechanism, using the same learnable parameters for both cross and self-attention but constructing the sequences from both the near field $- \mathcal{N}_\tau -$ and the points in $\beta_\tau$ respectively[8]. The new near-field operator, $\mathbf{G}_\theta(x_i, x_j)$ – represented in Figure 2 – is a local attention-based operator modelling the local contribution from equation 1. This is achieved by implementing a shared multi-head attention mechanism to compute interactions within the spatially local $3 \times 3$ neighbourhood. For each box, it gathers the features of the central box and their eight neighbours, performing two parallel attention passes: self-attention for within each box, and a cross-attention pass that incorporates the influence from neighbouring boxes. These outputs are added, and passed through an MLP and residual connections are applied – approximating the direct pairwise interaction kernel. We satisfy most of the requirements, as we mask-out self-interactions via a masked self-attention mechanism and cross-attention only computes the contribution from $\mathcal{N}_\sigma$ to $\beta_\tau$. However – as we do not currently use a relative position encoding scheme – we currently use RoPE (Su et al., 2021) in the local attention operator, however, using a relative position encoding scheme is a current focus of future work.

---

[7]As learning a fixed rank approximation to the Greens function $- G(x, y) -$ may cause an issue for oscillatory PDE problems.

[8]We split up the attention between the self and local contribution, this is done to prevent saturation of the Softmax within the attention scores.

# 4 NUMERICAL EXPERIMENTS

To evaluate the CloudNFMM against other neural operators, we utilised two time-harmonic datasets; PDEBench (KHOO et al., 2020), and WaveBench (Liu et al., 2024a). These datasets are both for time-harmonic PDE problems, these were used as the CloudNFMM is a direct solver and is in its present form not designed to solve time-dependent PDE problems.

The same hyperparameters were used for all the following numerical experiments on the Cloud-NFMM architecture. All scores in the tables below are the average relative $L_2$ error[9], $\mathcal{E}_2^{\text{rel}}$, values over the validation set. We trained the CloudNFMM using the average relative $L_2$ loss, $\mathcal{L}_2^{\text{rel}}$, for more information on training and implementation details see Appendix D.

## 4.1 RESULTS

### 4.1.1 WAVEBENCH

Table 1 shows the results of the CloudNFMM against the WaveBench benchmarks, which contain different Helmholtz problems. These results for the baseline architectures are referenced from the WaveBench paper, they have all been trained with the protocol outlined in Appendix D.

Table 1: Results of neural operators on the 2D acoustic Helmholtz datasets

| GRF Type | Freq. | CloudNFMM | FNO-depth-4 | FNO-depth-8 | U-Net-ch-64 | UNO-modes-16 |
|---|---|---|---|---|---|---|
| Isotropic | 10 Hz | **0.037** | 0.063 | 0.040 | 0.063 | 0.054 |
| | 15 Hz | 0.059 | 0.093 | **0.057** | 0.087 | 0.081 |
| | 20 Hz | **0.064** | 0.122 | 0.070 | 0.106 | 0.114 |
| | 40 Hz | **0.111** | 0.283 | 0.165 | 0.191 | 0.301 |
| Anisotropic | 10 Hz | 0.034 | 0.059 | **0.025** | 0.119 | 0.051 |
| | 15 Hz | 0.050 | 0.098 | **0.039** | 0.165 | 0.093 |
| | 20 Hz | 0.064 | 0.135 | **0.060** | 0.176 | 0.129 |
| | 40 Hz | **0.160** | 0.315 | 0.172 | 0.231 | 0.343 |
| **Total Parameters** | | **1.8M** | 4.2M | 8.4M | 31.0M | 17.9M |

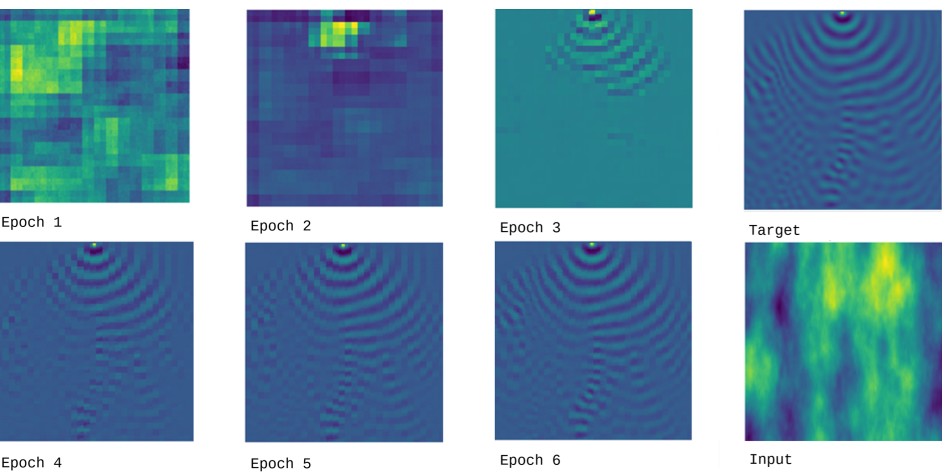

Figure 3: The first 6 epochs of the CloudNFMM model training on the 'Anisotropic 40Hz' dataset.

We present Figure 3, where we have visualised the first 6 epochs of the CloudNFMM model being trained on the 'Anisotropic 40Hz' dataset. We also conduct an experiment comparing training protocols and dataset sizes on the CloudNFMM, this is outlined in Appendix E.

---

[9]This is also occasionally called the normalised root mean squared error (nRMSE).

### 4.1.2 PDEBENCH

Table 2 shows the results of the CloudNFMM against the Darcy Flow benchmarks. These results for the baseline architectures are referenced from the HAMLET paper (Bryutkin et al., 2024), this was done so that we could best compare the CloudNFMM against other graph and transformer-based neural operators. They were all trained with the hyperparameters from the default implementation of baseline methods found within their respective papers[10], and trained and evaluated on $64 \times 64$ grids.

Table 2: Results of Neural Operators on the 2D Darcy Flow datasets.

| Darcy Flow $\beta$ | U-Net | FNO | DeepONet | OFormer | GeoFNO | HAMLET | CloudNFMM |
|---|---|---|---|---|---|---|---|
| $\beta = 0.01$ | 4.00e-03 | 8.00e-03 | 3.31e-03 | **2.21e-03** | 2.70e-03 | 2.45e-03 | 2.53e-01 |
| $\beta = 0.1$ | 4.80e-03 | 6.20e-03 | 4.88e-03 | 2.55e-03 | 4.15e-03 | **2.60e-03** | 1.06e-01 |
| $\beta = 1.0$ | 6.40e-03 | 1.20e-02 | 9.65e-03 | 3.00e-03 | 6.20e-03 | **2.74e-03** | 4.50e-02 |
| $\beta = 10.0$ | 1.40e-02 | 2.10e-02 | 6.79e-02 | 7.32e-03 | 2.08e-02 | **5.51e-03** | 1.39e-02 |
| $\beta = 100.0$ | 7.30e-02 | 1.10e-01 | 6.21e-01 | 4.91e-02 | 1.65e-01 | 3.37e-02 | **1.18e-02** |

We also conduct further experiments with the CloudNFMM, training the model on the $128 \times 128$ version of the Darcy flow dataset; these results are found in Table 7 in Appendix E.

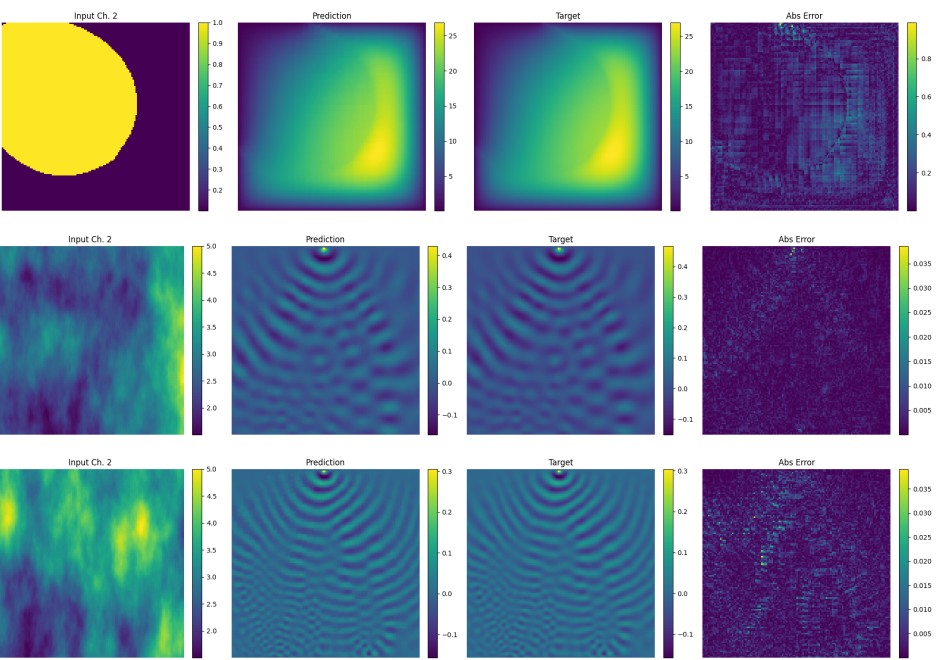

Figure 4: Examples from the validation set of: (*Top*) Darcy Flow $\beta = 100$, (*Middle*) Helmholtz Anisotropic 20Hz, and (*Bottom*) Helmholtz Anisotropic 40Hz

### 4.2 ANALYSIS

The results indicate that the CloudNFMM approach, outlined in this paper, is a potentially powerful performance-to-parameter efficiency class of neural operators. This is most apparent from the WaveBench dataset results – see Table 1, Figure 4, and Figure 3. Here, the model demonstrates exceptional performance, consistently outperforming most baseline models – particularly at higher

---

[10]The authors note that the, "...dataset-specific hyperparameters follow the PDEBench setting, while model-specific hyperparameters follow the default setting of baseline methods suggested by the code repositories or their papers.".

frequencies – while having the lowest parameter count (1.8M) by a significant margin. This suggests that the FMM-inspired hierarchical structure is highly effective at capturing the complex, this is supported by the first few epochs within training – as seen in Figure 3. Here the model quickly builds robust representations for far-field interactions using the hierarchical component, while the local transformer layer uses these to produce a smooth field. Conversely, the model's performance on the PDEBench Darcy Flow dataset – see Table 2 – is less competitive, especially for low to moderate values of the permeability coefficient $\beta$. Here the current CloudNFMM approach struggles compared to fully transformer-based operators – like OFormer and HAMLET.

### 4.3 LIMITATIONS AND NEXT STEPS

However, it is noteworthy that as $\beta$ increases to 100.0 – representing a very high-contrast problem – the CloudNFMM's performance improves to best-in-class. This result suggests that the current mechanism for exchanging information between the hierarchical FMM tree and the local attention operator may be a cause of the performance bottleneck. This suggests that the mechanism linking the coarse FMM tree and the fine-grained leaf interactions needs refinement. The current $\mathbf{T}^{\text{ofs}}$ and $\mathbf{T}^{\text{tfi}}$ operators might prevent local operator from building good representations for classes of PDEs with very smooth solutions.

Secondly, the current architecture is only formulated as a direct solver for time-harmonic problems[11]. We suggest addressing this using the same mechanism used by Bryutkin et al. (2024) in HAMLET, and originally outlined by Li et al. (2022a) for the OFormer – incorporating a recurrent structure in the latent space after the final layer of the model (before we apply $\mathcal{Q}$ in equation 2), which would propagate solutions through time.

Finally, while we deploy RoPE within the local attention layer, we aim to build representations which are informed by a learnt relative positional encoding. The absence of this information[12] may be preventing the CloudNFMM from learning fine-grained, geometry-dependent physical laws within the local neighbourhood.

Additional directions we aim to address in future work are: experimenting with replacing the MLPs in the CloudNFMM with SIREN (Sitzmann et al., 2020) – and KAN (Liu et al., 2024b) – models, and more comprehensive testing of the CloudNFMM on variable point densities and irregular samplings.

## 5 CONCLUSION

In this work, we introduced the Cloud Neural Fast Multipole Method (CloudNFMM), a discretisation-agnostic neural operator designed to solve time-harmonic PDEs on variable point densities (point cloud) data. We achieve this by adapting the information flow of the FMM, creating a hybrid method containing a hierarchical and local component. This allows our model to efficiently capture both long-range and short-range physical interactions without being constrained to a regular grid. Our experiments demonstrate that the CloudNFMM achieves state-of-the-art performance and remarkable parameter efficiency on challenging, highly oscillatory wave propagation problems. While its performance on smooth problems is an area of improvement in the coupling of local and global information flow, the results underscore the significant potential of this simple approach. The CloudNFMM represents a promising step towards creating scalable, discretisation-agnostic neural operators, indicating that fusion of principled numerical algorithms and deep learning architectures is a fruitful path for the future of scientific machine learning.

---

[11] This could be formulated as a fixed time horizon solver as well.

[12] This already included as an input to $\mathcal{P}$ in equation 2, but this indicates that this is a weak signal to the local attention operator.

ACKNOWLEDGMENTS

We acknowledge the use of Google's LLM, Gemini, as a writing assistant. The model was used to enhance grammar, improve phrasing, and improve the overall clarity of the text – but was not used for retrieval, discovery, or ideation. The authors carefully reviewed, revised, and take full responsibility for the scientific integrity and final content of this paper.

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

# A  THE FMM

The FMM is originally an efficient, hierarchical, numerical algorithm for computation of long-range forces in $N$-Body problems within gravitational and electrostatic fields developed by Rokhlin (1985) and has been extended by Ying et al. (2004) to apply the FMM to any Elliptic PDE with a Green's kernel. We will outline a high-level discussion of the FMM's information flow, for a deeper handling and derivation of the FMM we refer you to Martinsson (2019) for reference material. The FMM belongs to a family with linear or close to linear complexity for evaluating all pairwise interactions between $n$-particles, which is achieved by using two key ideas: a low-rank decomposition of the kernel, as seen in Figure 6, and hierarchically partitioning the spatial domain.[13] The FMM was originally designed to solve a $N$-Body interaction problems of the form equation 6, with $G(x, y)$ being the Green's kernel of the underlying physical problem, $x_i$ the set of point locations, $\phi_i$ the set of corresponding sources, and $u(x_i)$ being the set of potentials we wish to compute for all $1 \leq i \leq N$.

The functionality of the FMM stems from approximating far-field interactions using translation operators while directly computing only near-field interactions. We can best describe how these translation operators work together to compute a full level of the FMM by inspecting the operators needed to compute the interaction between two sufficiently separated boxes $\beta_\sigma$ and $\beta_\tau$. Here the $\tau$ and $\sigma$ subscripts indicate the target box and source box respectively, and 'sufficiently separated' means that $2b \leq \|c_\tau - c_\sigma\|$ with $c_\tau$ and $c_\sigma$ being the centre of $\beta_\tau$ and $\beta_\sigma$ respectively and $b$ being the length of a box at a given coarseness.

We denote $\mathcal{F}_\tau = \{\beta_\sigma | 2b \leq \|c_\tau - c_\sigma\|_1\}$ and $\mathcal{N}_\tau = \{\beta_\sigma | 2b > \|c_\tau - c_\sigma\|_1\}$ to be the *far-field* and *near-field* of $\beta_\tau$ respectively. We can compute $v_\tau$ from $\phi_\sigma$ by either a direct evaluation of $G(x, y)$ or compute it approximately by using the operators defined in equations equation 9 and equation 12.

$$u(x_i) = \sum_{j=1}^{N} G(x_i, x_j)\phi_j, \quad i = 1, 2, 3, ..., N \tag{6}$$

During the *upwards pass*, the sources $\phi_\sigma$ within a region $\beta_\sigma$ are translated into a single, compact outgoing vector, $q_\sigma \in \mathbb{R}^m$. Next, the *downward pass* maps this outgoing vector to a compact incoming vector, $h_\tau \in \mathbb{R}^m$, which is then propagated from the root down to the leaf level. Finally, the *leaf level pass* expands the far-field vector $h_\tau$ into approximate potentials $v_\tau$ and combines them with the direct evaluation of $G(x, y)$ for near-field particles.

## A.1  INTUITION FOR THE FMM ALGORITHM

We express the method by which the FMM separates computation between the local and far-field interactions via an analogy in terms of disjoint sets of our spatial domain, $D$. Consider a point $x \in D$, and the ball around $x$ of radius $r_0 - B(x, r_0)$ – we can partition our domain in the following way; $D = B(x, r_0) \sqcup B^c(x, r_0)$. Hierarchically decomposing $B^c(x, r_0)$ further, we can express $B^c(x, r)$ as a union of disjoint annuli: $B^c(x, r) = \cup_{i=1}^{\infty} A(x; r_{i-1}, r_i)$, where $r_{i+1} = 2 \cdot r_i$. Expressing this in terms of integrals with respect to our kernel, $G(x, y)$:

$$\int_D G(x, y)f(y)dy = \int_{B(x, r_0)} G(x, y)f(y)dy + \sum_{i=1}^{\infty} \int_{A(x; r_{i-1}, r_i)} G(x, y)f(y)dy \tag{7}$$

$$= \sum_{y_j \in \mathcal{N}_\tau} G(x, y_j)f(y_j)dy + \sum_{i=1}^{\infty} \sum_{y_j \in \mathcal{F}_\tau^i} G(x, y_j)f(y_j)dy \tag{8}$$

We arrive at equation 8 by describing equation 7 in-terms of sets of $\beta_i$. To achieve this, we identify $B(x, r_0)$ with $\mathcal{N}_\tau$, $b_l$ with $r_i - r_{i-1}$, and $A(x; r_{i-1}, r_i)$ with $\mathcal{F}_\tau^i = \{\beta_\sigma | 2b_l \leq |c_\tau - c_\sigma|\}$. As this paper is focused on modifying the computation of the near-field contribution, we will only outline the leaf pass.

---

[13]This is done via a Quadtree in $2D$ and an OctTree in $3D$.

## A.2 How to Construct a Quad-Tree

The FMM uses a quad-tree to hierarchically decompose a 2D domain, it is constructed in the following way:

- Let the total domain be represented by a unit square $[0, 1] \times [0, 1]$.
- This domain is recursively and uniformly subdivided down to a fixed depth, $d$.
- The number of patches, $M$, at this finest level, is given by $M = 4^d$.
- Each patch $m \in \{1, \ldots, M\}$ is a square region of side length $l = 1/2^d$.

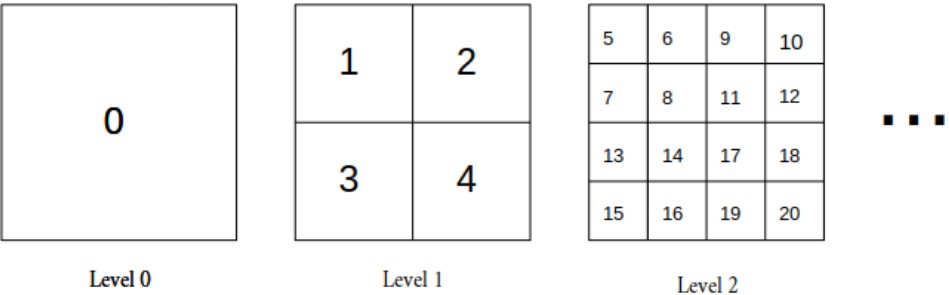

Figure 5: The Subdivision of the domain $\mathcal{D}$ into a QuadTree.

## A.3 Upwards pass

We begin with equation 9, which embeds the source terms into an outward potential in $\mathbb{R}^m$. To avoid repeated computation, this potential, $q_{\tau;l}$,[14] is translated from level $l$ in the tree, to level $l-1$. This is done by another operator,[15] $\mathbf{T}^{\text{ofo}}$, which combines the four child outward potentials into one outward potential for the parent box, $\beta_\Sigma$, with the potential centred at the centre of the parent box. This is mathematically represented in equation 10, letting $\mathcal{C}_\Sigma$ denote the children boxes of $\beta_\Sigma$.

$$q_\sigma = \mathbf{T}^{\text{ofs}}\left(\phi_\sigma\right) \tag{9}$$

$$q_\Sigma = \sum_{\tau \in \mathcal{C}_\Sigma} \mathbf{T}_l^{\text{ofo}}\left(q_\tau\right) \tag{10}$$

## A.4 Downward pass

Starting at level 2 and propagating down to the leaf level[16], denoted as level $l$, we combine potentials from the far-field. As moving down the quad-tree allows for finer spatial refinement, we can split up the incoming potential for a box, $h_{\tau,k} = h_{\tau,k}^\mathcal{P} + h_{\tau,k}^\mathcal{N}$, into two distinct components to reuse computation from the previous level. These two distinct components correspond to potential from the previous level of refinement, $h_{\tau,k}^\mathcal{P}$, and a component corresponding to the increased refinement from descending the tree, $h_{\tau,k}^\mathcal{N}$. To compute $h_{\tau,k}^\mathcal{N}$ we apply $\mathbf{T}_k^{\text{ifo}}$ to every sufficiently separated box not within the previous level of refinement, denoted $\mathcal{U}_\tau$. The incoming potential from the parent box corresponds to $h_{\tau,k}^\mathcal{P}$, this is shifted from the parent box, $\beta_T$, to the children boxes, $\beta_\tau$, by $\mathbf{T}^{\text{ifi}}$. We define $\mathcal{D}_k$ to be the downward pass for level $k$, which is mathematically represented for a single target box, $\beta_\tau$, in equation 11.

---

[14]The number corresponds to which level of the tree that the operator or vector corresponds to. For example $\mathbf{T}_2^{\text{ifo}}$ correspond to $\mathbf{T}^{\text{ifo}}$ on level 2 of the Tree.

[15]We simplify this process by only having one operator for $\mathbf{T}^{\text{ofo}}$ and $\mathbf{T}^{\text{ifi}}$.

[16]As the spatial resolution in the higher levels, levels 0 and 1, is too coarse to allow for separationn of the near-field and far-field.

$$h_\tau = \mathbf{T}_k^{\text{ifi}} h_T + \sum_{\sigma \in \mathcal{U}_\tau} \mathbf{T}_k^{\text{ifo}} q_\sigma \tag{11}$$

## A.5 LEAF LEVEL PASS

At the finest level of refinement, the leaf level, we are left to calculate the contribution from both $h_\tau$ and from the points in the near-field, $\mathcal{N}_\tau$. We apply $\mathbf{T}^{\text{tfi}}$ to expand $h_\tau$ into the far-field contribution of $v_\tau$. Those sources which lie within $\mathcal{N}_\tau$ we may compute by directly evaluating $G(x, y)$ between the sources in the near-field. This process is mathematically represented in equation 12.

$$v_\tau(x_i) = \mathbf{T}^{\text{tfi}}(h_\tau) + \sum_{\substack{j \in I_\tau \\ i \neq j}} G(x_i, x_j) + \sum_{\substack{\sigma \in \mathcal{N}_\tau \\ j \in I_\sigma}} G(x_i, x_j) \tag{12}$$

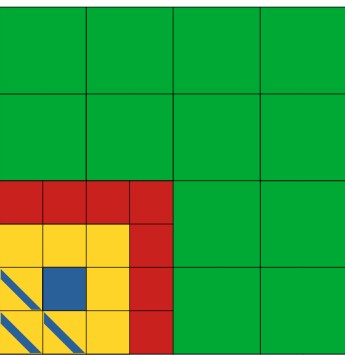

Figure 6: A visualisation of the decomposition of $\mathcal{D}$ about $\beta_\tau$, the blue square. Here we can see the different interaction sets of $\beta_\tau$, with the dashed blue and yellow boxes being the parent of $\beta_\tau$, $\beta_T$. The yellow boxes being $\mathcal{N}_\tau$, the green boxes being the far field of $\beta_T$, $\mathcal{F}_T$. The red and green (technically the children of the green boxes, but they define the same region) boxes being the far field of $\beta_\tau$, $\mathcal{F}_\tau$ and the red boxes being its unique far field, $\mathcal{U}_\tau$.

## B  THE NEURAL FMM

The efficiency of the FMM stems from approximating far-field interactions using translation operators while computing near-field interactions directly. However, a fundamental limitation of the traditional FMM is the requirement for an explicit, analytically available Green's kernel to derive these operators, which is difficult for problems in heterogeneous domains or where the kernel is unknown. Building upon the discussion of neural operators, our contribution is the **Neural Fast Multipole Method**, which integrates the information flow of the FMM while replacing the handcrafted, kernel-dependent translation operators with learnt operators parameterised by a learnable operators. We leverage the FMM's hierarchical partitioning and computation flow, outlined in equation 9 and equation 12, to split up and learn representations of local and far-field interactions. These passes are integrated into a single computational unit, the **Neural FMM Block**, with multiple of these blocks stacked together to form a deeper model, the **Deep Neural FMM**, enhancing expressivity of the model and mirroring equation 2.

### B.1  NEURAL FMM IMPLEMENTATION

The Neural FMM deviates from this by replacing each translation operator with an MLP while still following the non-local information flow for computing the far-field contribution, namely summing contributions from sufficiently separated boxes, rather than using a local operation at each level. The largest deviation from the FMM has been using one operator per level, $\mathbf{T}_{\theta;k}^{\text{ifo}}$, to represent the family

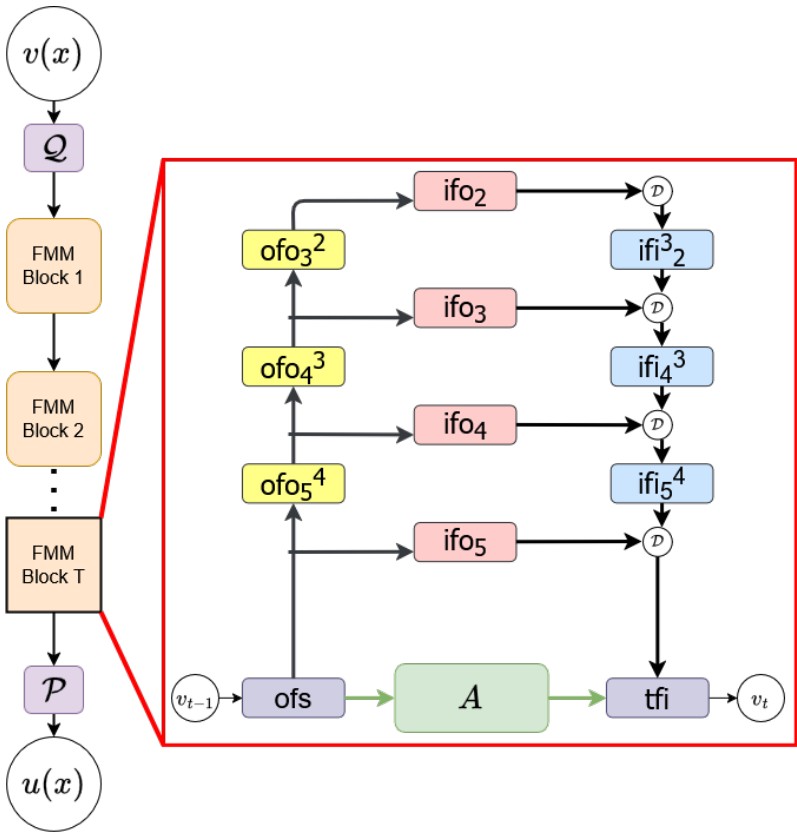

Figure 7: The Deep Neural FMM Architecture – We follow the neural operator framework from equation 2, with a lifting function $\mathcal{Q}$ and a projection function $\mathcal{P}$, while replacing each $\mathcal{K}_t(v_t))$ with a Neural FMM Block (outlined within the red box). Within the Neural FMM Block, $v_{t-1}$ passes up the Tree via the Upward Pass equation 9 and equation 10. This we then apply $\mathbf{T}_l^{\text{ifo}}$ to convert the outgoing potentials to their incoming potentials followed by propagating the information down the Tree via the Downward pass equation 11. The Leaf Pass then computes the contribution from the near-field, $\mathcal{N}_\tau$, using a MLP.

of linear maps which are derived from the translation function formula[17] from level $k$, represented by $\mathbf{T}_{\tau,\sigma}^{\text{ifo}}$; with a different matrix operator for each $\tau, \sigma$ pair.

### B.1.1   POSITION ENCODING

As the operators are applied channel-wise to every element in out domain at once, the network does not inherently know the spatial position of each element, which is core to how the Multipole-to-Local translation formula in the FMM performs the translation from outgoing potentials to incoming potentials. This required the inclusion of a spatial encoding scheme to reintroduce this spatial dependence for our MLP's. This was handled by the use of Rotary Position Embeddings (RoPE) (Su et al., 2021; 2022) applied to the vectors corresponding to each box, using the position of each box in the 1D Morton ordering as the position for RoPE. This approach was chosen as it was found to encode position information more directly, leading to better preservation of the spatial relationships between boxes when compared to additive sinusoidal encodings, and simpler to implement than a custom position encoding scheme.

### B.1.2   DOWNWARD PASS IMPLEMENTATION

The summations over interaction sets, particularly in the Downward Pass for the unique far-field $\mathcal{U}_\tau$ equation 11 is computationally intensive due to the non-local/non-contigous locations of the boxes within $\mathcal{U}_\tau$ with respect to the location of $\beta_\tau$. In order to increase the efficiency of the aggregation within the downward pass, we pre-compute masks corresponding to these interaction sets at the initialisation of the architecture.

## C   ATTENTION AS MESSAGE PASSING

Now that we have outlined that self-attention and cross-attention can be thought of as working on $K_N$ (a directed complete graph) or $\mathbf{K}_{N,M}$ (a complete undirected bipartite graph) respectively, we will now outline the corresponding form of:

- The Message Function: $K_\theta(\bullet, \bullet)$
- The Aggregation Function: $\square(\bullet, \cdots, \bullet)$
- The Update Function: $U(\bullet, \bullet)$

Without loss of generality, we will only consider the self-attention mechanism applied to a sequence of tokens $\{t_1, \ldots, t_N\}$ with input embeddings $\{x_1, \ldots, x_N\}$, as the only difference between cross and self attention is the topology of the underlying graph[18] while the mechanism itself is the same.

### C.1   MESSAGE FUNCTION

In a general MPNN, the message function $M_{vw}^{(t+1)} = M_t(x_v^t, x_w^t, e_{vw})$ computes a *message* dependent on the source node $w$, the target node $v$, and the edge features $e_{vw}$. Suppose we are on a graph without edge embeddings, so the adjacency matrix is a binary matrix indicating if nodes are connected together or not; the message function would then drop the dependence on $e_{vw}$. If we constrict the class of message functions to some kernel function, $K(\bullet, \bullet) : \mathbb{R}^D \times \mathbb{R}^D \to \mathbb{R}$. Within the attention mechanism, this $K$ is actually a bilinear product parameterised by two affine transformations, Key - $\mathbf{K}$ and Query - $\mathbf{Q}$, which means $K$ has the following form:

$$a_{i,j} = K(x_i, x_j) = (\mathbf{K}x_i)^T(\mathbf{Q}x_j) \tag{13}$$

### C.2   AGGREGATION FUNCTION

Since attention is a weighted sum, the aggregation function is summation, to produce the *context vector* $c_i$ for token/node $x_i$. However, within attention, there are two additional permutation invariant functions we apply. Firstly, we apply Softmax to the attention scores to compute a distribution,

---

[17]This is also called the Multipole-to-Local translation formula in the literature.

[18]However, assuming $\mathbf{S}$ or $\mathbf{T}$ are disjoint sets does allow for more efficient computation.

secondly we then apply the affine transformation - $\mathbf{V}$ - to the vector associated with the node/token, so the aggregation function has the following form:

$$c_i = \square_{x_j \in N(x_i)}(x_i) = \sum_{j \in N(x_i)} \left[ \frac{\exp(a_{i,j})}{\sum_{k \in N(x_i)} \exp(a_{i,k})} \mathbf{V} \right] x_j \tag{14}$$

$$c_i = \sum_{j \in N(x_i)} \left[ \frac{\exp\left(K(x_i, x_j)\right)}{\sum_{k \in N(x_i)} \exp\left(K(x_i, x_k)\right)} \mathbf{V} \right] x_j \tag{15}$$

$$c_i = \sum_{j \in N(x_i)} \left( \mathbf{Softmax}_{N(x_i)} \left[ K(x_i, x_j) \right] \right) \mathbf{V} x_j \tag{16}$$

Where $\mathbf{Softmax}_{N(x_i)}$ is Softmax normalised with respect to the neighbourhood of edges from $x_i$. As summation, Softmax, and matrix multiplication are permutation invariant, this is a valid message passing aggregation function.

### C.2.1 MULTIPLE HEADS

Multi-head attention does not change the core message passing framework, but instead paral-lelises the core logic over $h$ *heads*, which are subspace projections in which we perform atten-tion independently. Each head $h \in \{1, \ldots, H\}$ contain their own set of affine transformations, $(\mathbf{Q}_h, \mathbf{K}_h, \mathbf{V}_h) : \mathbb{R}^{\frac{D}{H}} \to \mathbb{R}^{\frac{D}{H}}$, although in practice one large affine transformation is used for the token sequence, which is then reshaped to match the number of heads. The individual head out-puts are concatenated to form a single, larger aggregated message, $c_i^H = c_i^{h_1} \| c_i^{h_2} \| \cdots \| c_i^{h_H}$. This concatenated vector is then passed through a final affine transformation, parameterised by $\mathbf{P}^O$, to produce the final output embedding:

$$c_i = \mathbf{P}^O \left( c_i^H \right) = \mathbf{P}^O \left( c_i^{h_1} \| c_i^{h_2} \| \cdots \| c_i^{h_H} \right) \tag{17}$$

In this view, multi-head attention is equivalent to computing multiple directional edges between each of the token/node embeddings, with the output of each of these subspace attention mechanisms being combined in the aggregation step.

$$K_h(x_i, x_j) = (x_i \mathbf{K})^T (\mathbf{Q} x_j)$$

$$c_i^h = \sum_{j \in N(x_i)} \left[ \frac{\exp\left(K_h(x_i, x_j)\right)}{\sum_{k \in N(x_i)} \exp\left(K_h(x_i, x_k)\right)} \mathbf{V}_h \right] x_j$$

$$c_i = \mathbf{P}^O \left( c_i^{h_1} \| c_i^{h_2} \| \cdots \| c_i^{h_H} \right)$$

Since the update function is applied to the context vectors $c_i$, multi-headed attention only modifies the message and aggregation function of the attention mechanism.

### C.3 UPDATE FUNCTION

Once we have computed the context vector $c_i$ via the message and aggregation function, we then need to update the representation of the token/node with respect to the context vector. In standard Transformer architectures the update function is a MLP with residual connections before and after the MLP, there are typically several normalisation operations[19], so the form of the update function is:

---

[19]These are typically either a pre-/post-application of LayerNorm between the residual connection and ap-plication of the MLP, but I have removed them for simplicity as they are added to improve stability during training.

$$U(x_i, c_i) = x_i + \mathbf{MLP}[x_i + c_i] \tag{18}$$

Thus, we can see that the attention mechanism implements a message passing scheme on a graph, where each weight is determined via the kernel message function $a_{ij} = K(x_i, x_j)$ - the query-key mechanism.

## D CLOUDNFMM IMPLEMENTATION DETAILS

### D.1 METRICS AND LOSS FUNCTIONS

#### D.1.1 RELATIVE $L_p$

The relative loss function – $\mathcal{L}_p^{\text{rel}}$ – computes a $l_p$ norm between the predicted and ground truth values, which is normalised by the $l_p$ norm of the ground truth – as originally outlined by N. Kovachki et al. (Kovachki et al., 2021b). For a predicted output tensor, $v$, and ground truth tensor, $u$, the relative error loss and relative error metric – $\mathcal{E}_p^{\text{rel}}$ – are computed as:

$$\mathcal{L}_p^{\text{rel}}(v, u) = \frac{1}{N} \sum_{i=1}^{N} \frac{\|v_i - u_i\|_p}{\|u_i\|_p} \tag{19} \qquad\qquad \mathcal{E}_p^{\text{rel}}(v, u) = \frac{\|v - u\|_p}{\|u\|_p} \tag{20}$$

where $p$ represents the order of the norm, and $N$ is the batch size. This loss provides a scale-invariant measure of error, particularly useful when dealing with solutions that may vary significantly in magnitude.

### D.2 TRAINING PROTOCOLS

#### D.2.1 DEFAULT TRAINING PROTOCOL

We found that training using Table 3 provided satisfactory results on a large class of problems.

Table 3: Default Training Protocol for the CloudNFMM

| # of Epochs | Optimiser | Scheduler | $lr_{\text{start}}$ | $lr_{\text{end}}$ | Train/Test Split |
|---|---|---|---|---|---|
| 50 | AdamW | OneCycle | 3e−4 | 3e−6 | 80/20 |

#### D.2.2 WAVEBENCH TRAINING PROTOCOL

This is the training protocol outlined in the WaveBench paper can be seen in Table 4.

Table 4: WaveBench Training Protocol

| # of Epochs | Optimiser | Scheduler | $lr_{\text{start}}$ | $lr_{\text{end}}$ | Train/Test Split |
|---|---|---|---|---|---|
| 50 | AdamW | Cosine Annealing | 1e−4 | 1e−6 | 99/1 |

**Note:** In the original WaveBench paper, they have $lr_{\text{start}} = 1\text{e}{-}3$ & $lr_{\text{end}} = 1\text{e}{-}5$. However, due to the training instabilities of transformer-based architectures at high learning rates, we reduced both $lr_{\text{start}}$ & $lr_{\text{end}}$ by an order of magnitude.

### D.3 MODEL HYPERPARAMETERS

We found that the CloudNFMM model hyperparameters outlined in Table 5 provided a good trade-off between architecture parameter size, model speed, and model performance. Using these model hyperparameters, we gained a model with 1.9M learnable parameters.

Table 5: Default CloudNFMM Architecture Hyperparameters

| Hyperparameter | Value | Description |
|---|---|---|
| *Dimensionality & Capacity* | | |
| source_dim | 3 | Dimensionality of the source data (e.g., $[x, y, f(x, y)]$). |
| target_dim | 1 | Dimensionality of the target data (e.g., $[u(x, y)]$). |
| leaf_dim | 64 | Feature dimension for leaf-level representations. |
| tree_dim | 128 | Feature dimension for tree node representations. |
| hidden_width | 256 | Width of hidden layers in network components. |
| root_rank | 256 | Operator rank at the FMM tree root. |
| leaf_rank | 256 | Operator rank at the FMM tree leaves. |
| *Core Architecture* | | |
| tree_depth | 5 | Depth of the FMM tree structure. |
| operator_depth | 2 | Depth of the operators in the neural operator blocks. |
| num_blocks | 4 | Number of stacked blocks in the main model. |
| *Regularization & Attention Details* | | |
| dropout | 0.1 | Dropout rate for regularization. |
| residual_connection | 'linear' | The kind of residual connection between blocks. |
| num_heads | None | Number of Attention Heads. |
| bias | True | Use a bias term in the MLPs. |
| use_rope | True | Use RoPE for the Attention mechanism. |

# E ADDITIONAL EXPERIMENTS

## E.1 WAVEBENCH

We also compare the effect of dataset size and training protocol on the performance of the Cloud-NFMM. For the 10k training dataset, we use the default training protocol for the NFMM, while for the 50k training dataset, we use the Wavebench training protocol – the results outlined in Table 6.

Table 6: Comparison of Dataset Size on CloudNFMM Isotropic and Anisotropic Results

| Dataset | 50k | 10k |
|---|---|---|
| Anisotropic 10Hz | **0.034** | 0.035 |
| Anisotropic 40Hz | 0.160 | **0.104** |

## E.2 PDEBENCH

We also conduct further experiments with the CloudNFMM, training the model on the $128 \times 128$ version of the Darcy flow dataset – the results outlined in Table 7

Table 7: Comparison of CloudNFMM trained on different resolutions

| Resolution | $\beta = 0.01$ | $\beta = 0.1$ | $\beta = 1.0$ | $\beta = 10.0$ | $\beta = 100.0$ |
|---|---|---|---|---|---|
| $64 \times 64$ | **2.53e-01** | **1.06e-01** | 4.50e-02 | 1.39e-02 | 1.18e-02 |
| $128 \times 128$ | 2.55e-01 | 1.25e-01 | 1.07e-01 | **9.91e-03** | **1.07e-02** |

