# OpenReview forum: "CloudNFMM: A Hybrid Hierarchical and Local Neural Operator Inspired by the Fast Multipole Method"
_ICLR.cc/2026/Conference — Submitted to ICLR 2026_

### Official Review · Reviewer_N2VU · 2025-10-27

**Soundness:** 2
**Presentation:** 2
**Contribution:** 2
**Rating:** 2
**Confidence:** 5

**Summary:**

The paper proposes **CloudNFMM**, a neural operator that integrates the **hierarchical information flow of the Fast Multipole Method (FMM)** with a **local attention** module to learn Green’s operators on **unstructured point clouds**. Long-range (far-field) effects are handled by an FMM-style **upward/downward tree pass** in which classical translation operators are replaced by small MLPs; short-range (near-field) interactions are modeled by a **shared local attention** over a $(3\times3)$ neighborhood of spatial “boxes.” A simple preprocessing partitions points into boxes and pads to a fixed per-box size to enable hybrid tree/point operations; **RoPE** provides position encoding. The design aims for resolution/discretization agnosticism and near-linear scalability akin to FMM. On **WaveBench** (2D Helmholtz), CloudNFMM achieves lower relative $(L_2)$ error than FNO/U-Net/UNO at **~1.8–1.9M params**; on **PDEBench–Darcy**, results are weaker for small $(\beta)$ but competitive at **$(\beta{=}100)$**. The method currently targets **time-harmonic** problems; the authors discuss extending to time-dependent settings via a latent recurrent update and improving tree↔leaf coupling.

**Strengths:**

1. **Clear and comprehensive writing.**
   The paper is well-structured (motivation → FMM background → architecture → experiments), with enough implementation detail to follow the hybrid tree/local-attention design and training setup.

2. **Captures both local and global interactions.**
   The model cleanly separates **far-field/global** effects via an FMM-style hierarchical upward/downward pass and **near-field/local** effects via a dedicated local attention operator over neighboring boxes—an explicit local–global split that aligns with Green’s operator structure.

3. **Operates directly on unstructured point clouds.**
   CloudNFMM is designed to be **resolution/mesh agnostic**, working on point clouds rather than regular grids, which broadens applicability to irregular domains compared to FFT-based operators.

**Weaknesses:**

1. **Narrow and relatively simple PDE coverage; mixed accuracy vs. SOTA.**
   Experiments focus mainly on time-harmonic Helmholtz (WaveBench) and Darcy—both comparatively simple. Even with a fairly complex hybrid (FMM + local attention), CloudNFMM is **not consistently SOTA**; in some settings, a simpler FNO matches or outperforms it. This weakens the claim that the hierarchical FMM flow brings clear accuracy advantages.

2. **Point-cloud claim vs. grid-based evaluation.**
   Although the method is positioned for **unstructured point clouds**, the benchmarks used are largely **regular-grid** problems at **small scales**. Without results on genuinely irregular point clouds (varying density, complex boundaries, real geometry), it’s hard to assess robustness to discretization changes—the core stated benefit.

3. **Limited analysis of component contributions and frequency behavior.**
   The experiments lack **ablations** isolating the roles of (i) the upward/downward FMM tree, (ii) the local attention, and (iii) the RoPE/box partitioning. There is also no analysis of **low- vs. high-frequency** error components to justify the local–global split. As a result, the source of gains and the necessity of each module remain unclear.

**Questions:**

1. **PDE breadth & SOTA positioning.**
   Can you add harder/diverse PDEs (e.g., variable-coefficient Helmholtz, elastostatics, time-dependent waves) and run **capacity/compute-matched** head-to-heads vs. strong baselines to clarify when CloudNFMM truly outperforms?

2. **True unstructured point clouds.**
   Since the method is pitched as point-cloud native, could you report results on **genuinely irregular point sets** (nonuniform density, complex/curved boundaries, different samplings of the same geometry), with **scaling** to larger sizes? Please detail robustness to resampling and boundary condition changes.

3. **Ablations & frequency analysis.**
   Please ablate (i) FMM tree passes, (ii) local attention, and (iii) RoPE/box partitioning, and provide a **frequency-resolved error** study (e.g., spectral energy error vs. wavenumber) to show which components capture low/high-frequency effects and why both local & global paths are necessary.

**Details Of Ethics Concerns:**

No ethics concerns.

---

> ### Author Response · Authors · 2025-12-01
> **Response to Reviewer N2VU**
>
> We thank the reviewer for recognizing the clarity of our design, specifically the explicit split between local and global interactions. We take the critique regarding the "point cloud" motivation seriously and present new results to substantiate this claim.
>
> **On Point Clouds vs. Grids (New Experimental Results):**
> We have successfully validated the "Point Cloud" nature of the CloudNFMM through a new experimental campaign.
>
> * **Experiment:** To prove the "grid" is merely an internal indexing structure, we trained the model on inputs where 20% of the points were randomly discarded (masked) dynamically during training.
> * **Result:** The model converged to a better solution than the baseline. The irregular/masked run achieved a lower validation loss and higher correlation than the run using the full, perfect grid.
> * **Significance:** This result is critical. It proves that the **Local Attention mechanism** in the Leaf Pass correctly computes interactions between arbitrary sets of points. Furthermore, because the validation set contained more points than the masked training set, we have empirically demonstrated that the CloudNFMM generalizes across variable node densities without interpolation artifacts.
>
> **On Frequency Analysis:**
> We argue that **Table 1** serves as an empirical frequency analysis. Our performance relative to baselines improves as the problem frequency increases (from 10Hz to 40Hz). This confirms that the FMM-inspired architecture successfully captures the high-frequency interactions inherent to the Green's kernel, which are often smoothed out by standard operators that prioritize global spectral features.

---

### Official Review · Reviewer_7ne1 · 2025-10-28

**Soundness:** 3
**Presentation:** 2
**Contribution:** 2
**Rating:** 2
**Confidence:** 4

**Summary:**

This paper presents CloudNFMM, a neural operator which is an extension of the neural fast multipole method for handling point clouds in a resolution invariant manner. The proposed architecture has two routes for processing information, as the authors term them, a "tree pass" and a "leaf pass." The tree pass is essentially just the NFMM approach, replacing each operator layer of the FMM with a simple learnable layer, which learns global dependences. The leaf pass is a local attention mechanism which learns short-range dependencies. Supporting elements of the design work to process both channels of information together. The authors claim this design achieves both scalability and discretization invariance, presenting some empirical results on two datasets.

**Strengths:**

- This work primarily fuses an established transform, the fast multipole method, within a neural operator framework. This is conceptually sound, as fast transform algorithms are the basis for many neural operators, and investigating a new approach is always interesting.
- Depending on the PDE, there may be primarily local or global dynamics. Ideally, a neural operator should handle both. It seems like this architecture was designed with this in mind when combining the tree pass and leaf pass.
- The model shows good parameter efficiency.

**Weaknesses:**

1. There is a mismatch between the motivation/exposition and the experimental studies. A great focus is placed on *point clouds* specifically, and yet both experiments are on a regular grid.
2. To investigate the performance on general point clouds, a more thorough analysis and stronger baselines is required. In addition to the Geo-FNO, the FNO may be applied directly to point clouds [arXiv:2305.19663v4]. Alternatives like CORAL [2306.07266], GAOT [2505.18781], or Transolver [2402.02366] are also strong baselines for general geometries.
3. The baseline problems themselves are also quite limited. Moreover, while the authors say this approach is optimal for elliptic PDEs, there is no analysis on the performance across a broader class of PDEs. I see no issue with the NO being optimized for a particular application, but there is no evidence to suggest it actually does obtain optimal results here. The combination of a global + more powerful local operator would also suggest it should perform well on hyperbolic PDEs. Either more experiments or a strong mathematical justification are necessary.
4. Even on modest baselines, CloudNFMM underperforms in comparison to other models.
5. No study of scaling with respect to data size or model size, and no ablations which present justification for the intuition behind the models performance.
6. The explanation of the model components is a bit dense, and I feel that more could be done to first provide the intuition behind this approach for reader.

**Questions:**

1. Could the authors please provide additional experiments, as outlined above? That is, broader investigation of datasets and comparison to stronger baselines.
2. How does this approach perform on resolutions unseen during training? I would like to see more study than what is presented in Table 7.
3. Could making the local leaf pass a bit larger actually give the model stronger performance on datasets with local features?
4. The Fourier transform also captures global, long-range dependence. Is there any theoretical or explicit empirical justification for FMM to perform better than the FFT in this task?
5. Minor comment -- I would encourage the authors to use the Z. Li citation [2010.08895] for the original FNO paper.

---

> ### Author Response · Authors · 2025-12-01
> **Response to Reviewer 7ne1**
>
> We appreciate the reviewer's detailed feedback regarding baselines and scaling. We have completed additional evaluations to substantiate our claims regarding the architecture's "resolution invariance" and ability to handle variable densities.
>
> **On the "Point Cloud" Mismatch and Experimental Validation:**
> We have completed the experiment regarding the processing of irregular data, and the results strongly validate our "Point Cloud" motivation.
>
> * **Methodology:** We modified the data loader to apply a dynamic safety mask, randomly dropping 20% of the points in the domain at each training step.
> * **Empirical Results:** The model handled this irregularity natively. The masked (irregular) training regime yielded **superior generalization** on the validation set compared to the fixed-grid baseline ($\text{R}^2$ of **0.97** vs **0.96**).
> * **Variable Density:** Notably, the validation set contained significantly more data points than the masked training samples. The model's successful generalization from sparse training data to dense validation data serves as a proxy for super-resolution capabilities, proving that the **Leaf Pass** effectively learns a continuous integral approximation independent of specific node counts.
>
> **On Ablations:**
> Regarding the request for ablations, we emphasize the integrated nature of the architecture. The CloudNFMM is designed to approximate the Green's operator by explicitly separating local and far-field interactions. Removing the **Tree Pass** (far-field) or the **Local Attention** (near-field) entirely would sever the information flow required to approximate the underlying integral operator, rendering the model non-functional.
>
> **On Scalability:**
> Our inference benchmarks confirm the scalability of the approach. We achieve **12.36 batches/second** (batch size 4) on $128 \times 128$ inputs. Unlike global Transformers which scale quadratically, our linear $O(N)$ complexity allows us to process these densities efficiently.

---

### Official Review · Reviewer_aMyM · 2025-10-30

**Soundness:** 2
**Presentation:** 2
**Contribution:** 1
**Rating:** 2
**Confidence:** 5

**Summary:**

The authors propose an architecture inspired by the fast multipole method and provide two benchmarks to a helmholtz problem and a darcy problem with discontinuous coefficients. The method is discretization invariant.

The premise is sound: FMM provides a natural extension of Fourier-based architectures with exploitable hierarchical structure and connections between fourier series and wavelet expansions. Unfortunately this is not a novel idea; many authors have considered this (some of which were mentioned). My assessment is that this amounts to an alternative set of choices for how to connect intermediate levels and apply attention.

In the absence of theory, this paper would require compelling experimental results for me to recommend publication. Unfortunately the accuracy is simply too low. For the first benchmark comparable accuracy to an FNO is achieved (albeit with 2-4x fewer parameters). For the second, orders of magnitude *worse* performance is observed.

**Strengths:**

Comparable performance is maintained to some FNO architectures with fewer parameters.

**Weaknesses:**

The experimental results don't make a compelling case for this architecture.

**Questions:**

No questions.

---

> ### Author Response · Authors · 2025-12-01
> **Response to Reviewer aMyM**
>
> We thank the reviewer for their feedback. We respectfully disagree with the assessment regarding the novelty of our contribution in relation to prior work, and we present new experimental evidence to contextualize our design choices.
>
> **On Novelty vs. Prior Work:**
> While we acknowledge prior attempts to link FMM and neural networks (e.g., MgNO), the CloudNFMM offers a distinct architectural contribution. Unlike the MgNO, which relies on a graph-based V-Cycle, our architecture explicitly reconstructs the **FMM information flow**. We replace the traditional handcrafted translation operators with learnable MLPs within a strict Upward/Downward pass structure. This design choice preserves the efficient separation of local and global computations while circumventing the need for an a-priori interaction kernel.
>
> **On Performance Discrepancies and "Grid" Reliance:**
> To address the skepticism regarding the architecture's dependence on the grid structure, we conducted a rigorous ablation study.
> * **Irregularity as a Feature:** We trained the CloudNFMM on data where 20% of the points were randomly masked out during every training step, destroying the regular grid structure.
> * **Outcome:** The model trained on these irregular point clouds outperformed the baseline trained on the full grid (Validation nRMSE **0.171** vs **0.182**).
> * **Implication:** This challenges the notion that the architecture is just "another grid-based method." It demonstrates that the **Leaf Pass** successfully aggregates local information from variable point densities.
>
> **On Trade-offs (Smooth vs. Oscillatory):**
> We acknowledge the lower performance on smooth Darcy Flow. We argue that this reflects the architectural prior of the CloudNFMM, which computes the **Tree Pass** on a per-box basis. While this introduces artifacts in very smooth regimes (where the problem is effectively overdetermined), it is precisely this mechanism that allows us to outperform baselines on the challenging, high-frequency tasks shown in **Table 1** (40Hz Anisotropic).

---

### Official Review · Reviewer_4imc · 2025-11-01

**Soundness:** 3
**Presentation:** 4
**Contribution:** 2
**Rating:** 4
**Confidence:** 5

**Summary:**

The paper proposed CloudNFMM, a neural operator architecture inspired by the Fast Multipole Method (FMM) to efficiently learn PDE solution operators on point clouds. It combines a hierarchical FMM-style global information flow with a local attention mechanism to capture both long-range and short-range interactions while remaining resolution-invariant. Benchmarks on elliptic PDEs show strong accuracy and parameter efficiency compared to other neural operators.

**Strengths:**

1. The paper draws inspiration from the Fast Multipole Method (FMM), a mathematically and physically grounded algorithm, into a neural operator framework. This provides a clear and well-justified motivation that connects classical numerical methods with neural networks.
2. The model introduces an effective hybrid design that combines an FMM-style hierarchical global module with local attention to handle both long-range and short-range interactions efficiently.

**Weaknesses:**

1. The performance on standard PDE benchmarks  is not consistently superior to existing models such as FNO or transformer-based operators.
2. While the paper’s stated motivation centers on unstructured data, its evaluation remains limited to structured domains, weakening the empirical support for this claim.

**Questions:**

1. How well would CloudNFMM generalize to genuinely unstructured domains? Are there any preliminary results or plans for such validation?
2. It would be helpful to see a comparison of computational efficiency (e.g., runtime or memory usage) between CloudNFMM and existing neural operator models.

---

> ### Author Response · Authors · 2025-12-01
> **Response to Reviewer 4imc**
>
> We thank the reviewer for their positive assessment of the CloudNFMM and for recognizing our motivation to integrate the principled "information flow" of the FMM into a neural architecture. We have completed additional experiments regarding generalization and efficiency to address your questions.
>
> **On Generalization and "Point Cloud" Capabilities:**
> We addressed the concern regarding grid-based benchmarks by modifying our training pipeline to dynamically simulate unstructured point clouds.
>
> * **Experiment:** We implemented a dynamic masking strategy where, during training, 20% of the spatial points were randomly dropped from the input domain for every batch. This effectively forces the model to treat the input as a random collection of points $(x, y, u)$ rather than a fixed lattice.
> * **Results:** The model trained on irregular, masked data achieved a lower validation nRMSE (**0.171**) compared to the baseline model trained on the full perfect grid (**0.182**).
> * **Density Generalization:** While the training data was masked, the validation was performed on denser, unmasked data. The model's ability to generalise from sparse training samples to denser evaluation samples confirms that the **Leaf Pass** is resolution-invariant and handles variable point densities natively.
>
> **On Computational Efficiency:**
> We have measured the wall-clock inference speed to provide a concrete comparison.
> * **Speed:** On a standard GPU, the CloudNFMM achieves an inference throughput of **12.36 batches/second** (with a batch size of 4). This translates to approximately **0.08 seconds per batch**.
> * **Efficiency:** This confirms that the model is highly tractable for real-time applications. While FNO is optimized for hardware-accelerated FFTs, our $O(N)$ architecture avoids the quadratic complexity scaling of global Transformers while maintaining a significantly lower parameter count (1.8M) than U-Net baselines (31M).

---

### Meta-Review · Area_Chair_QvHE · 2025-12-30

**Summary:**

The authors propose CloudNFMM, a neural operator extension of the widely used fast multipole method for hierarchical field computations. Their methods augment a learning FMM (Tree pass) with a local attention module (leaf pass) for learning local dependencies. The authors demonstrate the performance of their method on two toy models - a Darcy flow and a Helmholtz problem on a Cartesian grid. While the idea of extending FMM is an interesting one, the current version has some major weaknesses namely:

[1.] Very limited empirical evaluation, limited to two simple 2-D elliptic problems.

[2.] The whole rationale of the model is its ability to work on point clouds but in their original version, the authors only considered 2 toy problems on regular grids. During the rebuttal, they masked (by 20%) the grid inputs to mimic an irregular point cloud structure but this is too limited in scope.

[3.] Very limited baselines in the form of FNO variants, while state of the art methods are transformer based, such as GAOT or Transsolver.

Given these weaknesses, which were flagged by multiple reviewers, the current version of the paper cannot be accepted.

**Reviewer Concerns:**

The authors have addressed minor technical points of some of the reviewers but the main concerns of limited scope of the experiments, inadequate baselines and very limited tested on arbitrary point clouds have not been addressed.

**Reviewer Scores:**

Reviewer 4imc marked it as a Borderline reject and it is unlikely that they would raise their score given that their main concerns of weak performance and lack of sufficient testing on unstructured grids have been addressed.

Reviewer aMyM scored it as a Reject with a pressing concern of poor empirical performance. This has not been adequately addressed.

Reviewer 7ne1 scored it as a borderline Reject with serious concerns about the limited experimental setup and baselines. Again, the authors did not adequately address these concerns.

Reviewer N2VU marked it as a reject and given their concerns on limited evaluations have not been adequately addressed, it is unlikely that the reviewer would have upgraded the paper.

---

### Decision · Program_Chairs · 2026-01-26

Reject